**Data Availability Statement:** Data cannot be made public because it describes participant experiences which are not de-identifiable and consent for release was not provided by study participants. The

# Implementation of a volunteer contact tracing program for COVID-19 in the United States: A qualitative focus group study

Tyler Shelby[1,2☯], Rachel Hennein[1,2☯], Christopher Schenck[2], Katie Clark[1], Amanda J. Meyer[1], Justin Goodwin[1,2], Brian Weeks[3¤], Maritza Bond[3], Linda Niccolai[1], J. Lucian Davis[1,4,5]*, Lauretta E. Grau[1]

1 Department of Epidemiology of Microbial Diseases, Yale School of Public Health, New Haven, Connecticut, United States of America, 2 Yale School of Medicine, New Haven, Connecticut, United States of America, 3 New Haven Health Department, New Haven, Connecticut, United States of America, 4 Pulmonary, Critical Care, and Sleep Medicine Section, Yale School of Medicine, New Haven, Connecticut, United States of America, 5 Center for Methods in Implementation and Prevention Science, Yale School of Public Health, New Haven, Connecticut, United States of America

☯ These authors contributed equally to this work.
¤ Current address: Norwalk Health Department, Norwalk, Connecticut, United States of America
* Lucian.Davis@yale.edu

## Abstract

### Background

Contact tracing is an important tool for suppressing COVID-19 but has been difficult to adapt to the conditions of a public health emergency. This study explored the experiences and perspectives of volunteer contact tracers in order to identify facilitators, challenges, and novel solutions for implementing COVID-19 contact tracing.

### Methods

As part of a study to evaluate an emergently established volunteer contact tracing program for COVID-19 in New Haven, Connecticut, April-June 2020, we conducted focus groups with 36 volunteer contact tracers, thematically analyzed the data, and synthesized the findings using the RE-AIM implementation framework.

### Results

To successfully *reach* cases and contacts, participants recommended identifying clients' outreach preferences, engaging clients authentically, and addressing sources of mistrust. Participants felt that the *effectiveness* of successful isolation and quarantine was contingent on minimizing delays in reaching clients and on systematically assessing and addressing their nutritional, financial, and housing needs. They felt that successful *adoption* of a volunteer-driven contact tracing model depended on the ability to recruit self-motivated contact tracers and provide rapid training and consistent, supportive supervision. Participants noted that *implementation* could be enhanced with better management tools, such as more engaging interview scripts, user-friendly data management software, and protocols for special

New Haven Health Department provides oversight for the data collected by and for its contact tracing program, and by policy requires any use of this data to be directly approved by the Health Department. For these reasons, data may only be made available upon request made to the Corresponding Author and the New Haven Health Department. Please direct data requests to the following non-author email at the Yale School of Public Health's Department of Epidemiology of Microbial Diseases: kimberly.rogers@yale.edu.

**Funding:** This work was supported by a grant from the Dean's Office at the Yale School of Public Health. The funders had no role in study design, data collection and analysis, decision to publish, or preparation of the manuscript.

**Competing interests:** JLD, AJM and TS declare a contract with the state of Connecticut to assist with the state's contact tracing program. This does not alter our adherence to PLOS ONE policies on sharing data and materials.

situations and populations. They also emphasized the value of coordinating outreach efforts with other involved providers and agencies. Finally, they believed that long-term *maintenance* of a volunteer-driven program requires monetary or educational incentives to sustain participation.

## Conclusions

This is one of the first studies to qualitatively examine implementation of a volunteer-run COVID-19 contact tracing program. Participants identified facilitators, barriers, and potential solutions for improving implementation of COVID-19 contact tracing in this context. These included standardized communication skills training, supportive supervision, and peer networking to improve implementation, as well as greater cooperation with outside agencies, flexible scheduling, and volunteer incentives to promote sustainability.

## Background

The arrival of the coronavirus disease 2019 (COVID-19) pandemic in the US will be long remembered for its suddenness and severity. In the first six months following its arrival in the US [1], there were over three million cases and over 100,000 deaths [2]. As vaccine hesitancy and new viral variants raise the possibility that COVID-19 will become an endemic disease, contact tracing will continue to play a critical role in suppressing local epidemics and lessen the need for stay-at-home orders or other forms of strict physical distancing restrictions. However, the magnitude of the COVID-19 crisis and rate of its spread throughout the US has posed a challenge to implementing contact tracing at the required scale [3]. The high reproductive number [4], lengthy incubation period [5], frequency of pre-symptomatic transmission [6, 7], occurrence of super-spreader events [8], and large proportion of asymptomatic cases [9] have set COVID-19 apart from most other infectious diseases for which contact tracing is used, such as foodborne illnesses [10, 11], sexually transmitted infections [12], tuberculosis [13], and others [14–16]. These, the defining characteristics of the COVID-19 pandemic, have ensured that there would be huge numbers of cases and contacts and a resulting need for extremely large tracing workforces to investigate exposures and interrupt the many chains of transmission [17, 18]. Meanwhile, the lack of feasibility and acceptability of the best alternative, digital contact tracing, has ensured that person-led strategies will likely remain the first-line approach in most settings [19].

In response to these challenges, many states and local health departments rapidly expanded capacity for COVID-19 contact tracing early in the pandemic [20]. Massachusetts, Ohio, Indiana, and Maryland partnered with vendors to facilitate the hiring and management of thousands of new contact tracers, while Washington, Alabama, California, and Tennessee reassigned state employees to this role. Some states such as Rhode Island, West Virginia, North Dakota, and Washington activated their National Guard, and still other states engaged volunteers to fill the role of contact tracers, including Oklahoma, Kansas, Michigan, Arizona, and Connecticut. Learning from prior efforts is paramount given the continued role that contact tracing will play in helping us exit the pandemic.

Implementation science frameworks can aid in systematically identifying and understanding the relationships between factors that influence implementation successes and failures. The RE-AIM framework has been employed extensively for this purpose [21, 22] and contains five dimensions: (1) *reach*, which focuses on the population an intervention targets and the

process of engaging them, (2) *effectiveness*, which focuses on the intended impact of an intervention and potential barriers to that impact, (3) *adoption*, which focuses on the setting and individuals delivering the intervention, (4) *implementation*, which focuses on intervention protocols and strategy, and (5) *maintenance*, which focuses on intervention sustainability and scalability. Because volunteers were and still are key stakeholders in many contact tracing programs, learning about their experiences is vital for sustaining and scaling up contact tracing. To this end, we conducted focus group discussions (FGDs) with volunteers participating in a contact tracing program in Connecticut. We sought to characterize their perspectives and experiences using the RE-AIM framework in order to understand facilitators of, barriers to, and potential solutions for improving implementation.

## Methods

This qualitative study was part of a larger multiple methods evaluation of a volunteer-driven contact tracing program established in a partnership between the New Haven Health Department, hereafter referred to as the "Health Department", and Yale School of Public Health (YSPH) in March 2020. We report our methods below in accordance with the Consolidated Criteria for Reporting Qualitative Research (CO-REQ, see S1 Table).

### Setting and procedures

New Haven is home to nearly 130,000 residents and is part of the New York Metropolitan area. The Health Department established a partnership with Yale University for volunteer contact tracing on March 27, 2020, as previously described [23]. Briefly, over 150 volunteer students, staff, and faculty from Yale's public health, medical, physician assistant, and nursing programs participated in the program. Volunteers began making contact tracing calls on April 4, 2020, prior to New Haven's initial peak of COVID-19 cases around April 21 [24]. In mid-April, the Health Department assigned 40 public health nurses to assist with contact tracing. By mid-May, the program had responded to over 2,000 lab-confirmed cases of COVID-19.

Volunteers worked remotely and were divided into two teams. One team ("case investigators") interviewed cases to identify contacts and counsel self-isolation, while the other team ("contact notifiers") notified contacts about their exposure to COVID-19 and recommended self-quarantine. The case investigation team was supervised jointly by the Health Department and YSPH, and the contact notification team was supervised by YSPH faculty and staff. Volunteers participated in a one-hour, virtual training session on contact tracing that covered US Centers for Disease Control and Prevention (CDC) guidelines [25], local case- or contact-specific protocols for implementing contact tracing, and regulations for protecting confidentiality. All case investigator volunteers received training on basic communication and interviewing skills, except medical students who all had prior training in this area. Volunteers used email and GroupMe (Microsoft, New York, NY), a mobile group chat application that hosts discussion threads, to communicate with supervisors or other team members as needed.

Each day, the Health Department's lead epidemiologist identified new positive COVID-19 cases from the state's reportable disease database and shared their corresponding outreach information with the case investigation team. Case investigators used New Haven's existing emergency management software (Veoci, New Haven, CT) to record call attempts and responses to the interview questions. The case investigator team shared a daily list of reported contacts, without any information regarding their respective cases, with the contact notification team via email. This team then used a free-text template (Microsoft Word, Redmond, WA) to record notes and outcomes of call attempts to contacts. These data were then entered into a master spreadsheet (Microsoft Excel, Redmond, WA) by volunteers assigned to data

management tasks. Case investigators routinely asked cases about food or housing insecurities, ability to isolate within homes, access to medical care, and other social needs, while providing numbers to local support organizations or free clinics when applicable. Contact notifiers also provided links to resources when applicable but did not routinely assess contacts for the same needs. Team leads communicated changes in guidelines and protocols to volunteers via email and modified data collection forms appropriately.

## Eligibility and recruitment of volunteers

Eligibility criteria included being a volunteer in the case investigation or contact notification teams. We excluded the less experienced case investigators, defined as being in the lowest 25th percentile of total case assignments (<7 assignments). We did not exclude any contact notifiers because all assignments were distributed equally among this team, whereas case investigators were able to adjust their availability each week. We emailed invitations to all eligible volunteers to participate in the study. We set an initial recruitment goal of 18 participants from each team based on estimates of the number of focus groups required for thematic saturation [26]. We enrolled participants consecutively until the target sample size was reached, ensuring balanced representation of volunteers from different schools and university positions (*i.e.*, students, faculty, and staff).

## Data collection

Three members of the research team (TS, a male MD/PhD student; KC, a female research associate with a master's degree in public health; LG, a female social scientist and faculty researcher with a doctorate in psychology) conducted the focus groups. KC and LG led the case investigation discussions as moderator and scribe, respectively. TS and LG led the contact notification discussions, each serving as moderator or scribe. All had previous training or experience in conducting qualitative interviews. Because several participants knew TS as a fellow student and volunteer assistant coordinator of the case investigation team, he participated in the contact notification FGDs only. All participants were informed at the start of the discussions of the researchers' role in evaluating the volunteer contact tracing program. The FGDs were held via videoconferencing (Zoom, San Jose, CA) and conducted separately for case investigators and contact notifiers. The semi-structured FGD guide, developed around our primary purpose statement, included four domains: 1) experiences volunteering with the program, 2) successes and challenges related to contact tracing activities, 3) training and unforeseen experiences, and 4) perspectives on how to improve and sustain the program. After each FGD, participants received a follow-up survey inviting them to provide demographic information and any additional thoughts or comments they had.

We transcribed session recordings using an automated transcription service (Trint, London, United Kingdom). Additional researchers (AM, RH, CS) reviewed transcripts for accuracy against the audio and video recordings. Two moderators (TS and LG) iteratively assessed the content of case investigator sessions until no new themes emerged (*i.e.*, saturation had been reached), and separately followed the same process for contact notifier sessions [26]. We did not conduct follow-up interviews or discussions and did not have participants review the transcripts.

## Analysis

The coding team (TS, RH, LG) independently reviewed one case investigator and one contact notifier transcript and met to discuss and develop the codebook inductively. They discussed and resolved all coding discrepancies by consensus. Once acceptable inter-coder agreement

[27] was reached, TS and RH divided the remaining transcripts and free-text responses from the follow-up surveys between themselves for independent coding. The full coding team continued to meet regularly to resolve any remaining coding questions. The coding team initially used Microsoft Word for coding, and the data were subsequently entered into ATLAS.ti (Version 8, Berlin, Germany) and analyzed iteratively using thematic analysis [28]. Study participants did not provide feedback on the findings.

After the themes had been identified, we used the RE-AIM framework [21, 22] to deductively organize emergent themes. We assigned themes related to contacting and engaging clients to the *reach* dimension, challenges to achieving public health outcomes (*i.e.*, isolation for cases and quarantine for contacts) to the *effectiveness* dimension, volunteer delivery of the intervention and the setting in which they operated to the *adoption* dimension, and feasibility and acceptability of the program to the *implementation* dimension. The final theme concerning the sustainability of a volunteer-driven contact tracing program was assigned to the *maintenance* dimension. Once organized according to the RE-AIM framework, we identified specific barriers, facilitators, and solutions within each RE-AIM dimension.

### Ethics statement and consent procedures

The study protocol was approved by the Yale Human Subjects Committee (Institutional Review Board Panel A for Social, Behavioral, and Educational Research) and the New Haven Health Department. A waiver of written consent was approved by the Human Subjects Committee because the study posed no greater than minimal risk and did not involve any procedures that would require written consent in a non-research context. Before video-recording the session, the group facilitators read the consent form aloud and obtained verbal consent from all participants to be in the study and be recorded.

## Results

### Characteristics of the study sample

At the time of study recruitment, there were 106 case investigation volunteers and 36 contact notification volunteers involved in the program. We emailed 83 eligible volunteers from the case investigation team and 36 from the contact notification team, excluding 23 case investigators who made too few calls. We consecutively enrolled all participants who replied to the initial recruitment emails, sending reminder emails until we recruited a sample of 18 participants from each group. The six FGDs (three with case investigators and three with contact notifiers) ranged from 73 to 85 minutes in duration and occurred May 6–12, 2020. Six participants attended each session. Table 1 describes the sample characteristics. School affiliations within the sample were similar to the those on the volunteer team overall, with a slightly lower representation of nursing students and a higher representation of faculty and staff in the study sample.

### Identified themes

We identified 12 themes across the five RE-AIM dimensions. There were no differences in themes expressed by volunteer type or by participant demographics or between the FGDs and follow-up free-text surveys.

### Reach dimension

We identified two themes, Making Contact and Establishing Rapport, under the *reach* dimension. These captured volunteers' experiences attempting to get in touch with and engage the target population.

**Table 1. Participant characteristics.**

| Characteristics | Case Investigators (n = 16) [*] | Contact Notifiers (n = 17) [*] |
|---|---|---|
| | n (%)[†] | n (%)[†] |
| Age, median years (Q1, Q3) [§] | 28 (27, 29) [§] | 25 (22, 28) [§] |
| Female | 12 (75) | 14 (82) |
| Race/Ethnicity | | |
| Non-Hispanic White | 12 (75) | 13 (76) |
| Asian | 3 (19) | 1 (5.9) |
| Hispanic/Latinx | 1 (6.3) | 3 (18) |
| University Affiliation[*] | | |
| Public Health Student | 6 (33) | 16 (89) |
| Medical Student | 9 (50) | 0 (0) |
| Nursing Student | 1 (5.6) | 0 (0) |
| Post-graduate | 1 (5.6) | 0 (0) |
| Faculty / Staff | 1 (5.6) | 2 (11) |
| Bilingual[¶] | 3 (0.19) | 3 (0.18) |

[*] Only 33 of the participants completed the follow-up demographic surveys, thus demographic and language information about three participants is not included in this table. University affiliation was available for all participants.

[†] Unless otherwise specified.

[§] Median (quartiles 1 and 3).

[¶] Conducted interviews/notifications in Spanish in addition to English.

**Making contact theme.** Participants detailed their experiences calling and attempting to reach cases and contacts and described the challenges they faced with this early step of the contact tracing process. They noted the difficulty in getting the target individuals to answer their calls and reported that it was rare for their unanswered calls to be returned. However, some succeeded by either leaving voicemails or using text messaging in addition to voicemails.

*"The hard part is getting them on the phone in the first place, to answer the phone or return the voicemail." (Participant 1:6, Case Investigator)*

*"When they don't have [a voice mailbox], I've just been sending them a text with information from the callback scripts. I don't know whether that is appropriate or inappropriate, but I felt that that would be how I would want to get the information." (Participant 5:1, Contact Notifier)*

Several also noted that calls made in the afternoon or evening were more likely to be answered than those made in the morning.

*"I found that a lot of cases don't like being called in the morning. As I started, I would call at 9:00 or 10:00 in the morning, cause I just felt like maybe that would be a reliable time to get people and was also convenient for me. More than once I was basically told [by cases] 'don't call before noon.' So, I no longer call before noon." (Participant 1:5, Case Investigator)*

**Establishing rapport theme.** Volunteers repeatedly emphasized the importance of engaging with clients authentically. Participants felt that finding the most convenient times for the conversation, showing empathy, and addressing sources of mistrust were effective in building trust and rapport. Some volunteers developed these approaches by drawing on prior experiences in patient care or other client-related work, while others did so by trial and error.

"*I've personally gotten a few of those calls where they don't appreciate the call. They don't want to talk to you. It's been interesting [figuring out] how exactly do you handle those, because at first, I was really nervous making those calls but now it's been a lot more natural and it's been a very interesting process, kind of learning how to do that.*" (Participant 6:5, Contact Notifier)

"*I think that's why it's so critical to have been in the health care profession beforehand, because a lot of these questions are very sensitive, and you have to kind of know how to deal with that and make it okay. So, I would say yeah, training through med school has helped.*" (Participant 1:2; Case Investigator)

Some call recipients seemed suspicious of callers, and participants occasionally felt "awkward" trying to convince these individuals that they were authorized representatives of the Health Department. Others described the process of eliciting information about contacts from cases as particularly difficult because many cases either felt uncomfortable providing or simply did not know the necessary information about their contacts. One participant stated that a few cases disclosed their status as undocumented immigrants and were fearful about providing information about themselves or their contacts. Despite these challenges, participants stated that most cases and contacts appeared to be "very receptive" to providing information and following the recommended guidelines.

"*People are very guarded about who's in their house. . .But I think half the time it's the person. They're just a little bit wary. And half the time it's just the situation. Like, they would love to tell you, but they're also scared. And the other portion of the time people are just really open and they're trusting and then it's not a big deal.*" (Participant 1:2, Case Investigator)

## Effectiveness dimension

We identified two themes within the *effectiveness* dimension, Delays and Community Needs. Both concerned barriers to achieving the desired outcomes of isolation for cases and quarantine for contacts.

**Delays theme.**   Participants discussed several types of delays that prevented them from reaching cases and contacts within an epidemiologically relevant timeframe. There were delays in receiving test results and delays when a volunteer could not speak the client's preferred language, requiring reassignment to a volunteer proficient in the preferred language on the following day. These delays sometimes resulted in reaching contacts after the two-week window for effective quarantine had expired. Others described the frustration of reaching contacts only to discover that they had already been diagnosed with COVID-19.

"*We have no idea when things are getting reported to the state, when the state then goes to the city, when the city forwards that result along to our coordinators, and then when they finally put it on our list. . .there are some health clinics that seem to be slower reporters.*" (Participant 1:5, Case Investigator)

While some delays in the overall contact tracing process were beyond the control of the program, such as cases choosing to delay seeking COVID testing or slow reporting of test results, participants felt that identifying cases in need of translators before the first call was an actionable way to prevent additional delay.

 

"*I know that they're pulling the data from the state database but having a flag for language would really cut down in terms of time, because we're talking about an extra 24 hours.*" *(Participant 4:1, Case Investigator)*

**Community needs theme.**   Even when reached in time, participants stated that many cases and contacts indicated that they were either experiencing or expecting difficulties in adhering to isolation or quarantine recommendations. These challenges stemmed from job or wage loss, difficulties providing food for themselves or their families, and for some, a lack of housing. Participants observed that these challenges occurred more frequently among contacts from Hispanic communities and that contact tracing calls provided a unique opportunity to identify additional needs for support or resources.

"*I had one case or contact that I called, and they said, 'There's no one else in my house who can get groceries. I'm the only one who can go out. I don't know how we're going to get food. My husband is very sick and I'm trying to take care of him.'*" *(Participant 4:5, Contact Notifier)*

"*When I speak to Spanish-[speaking] contacts. . .what I hear more often is, 'I can't not go to work.' And I don't hear that as much when I [call] other contacts that I receive. I just hear people panicking essentially over the phone.*" *(Participant 5:5, Contact Notifier)*

Participants appreciated that the Health Department instructed volunteers to routinely assess these needs and thought many cases and contacts viewed this needs assessment as a sign of the city's concern for its residents. However, some participants were unsure whether these needs would ultimately be addressed by the city and questioned the utility of assessing needs when they could offer cases and contacts little assurance that the needs would be satisfied.

"*The other thing that was added actually fairly recently was [a prompt asking if] they have a need for housing support, food support, financial assistance, any of those things. . .it just adds a human element to the interview because, by asking that, it shows that we're not only doing this to use the participants as a source of information, but also we're here because we care about them as part of the community. So, it adds that element that I think people are very receptive to.*" *(Participant 2:2, Case Investigator)*

## Adoption dimension

We identified five themes that fit under the *adoption* dimension: Volunteer Motivations, Time Management, Knowledge, Skills, and Collaborative Learning. They addressed the dimension's focus on the individuals delivering the intervention (*e.g.*, what their motivations were for volunteering); the time, knowledge and skills required of those individuals; and key characteristics of the setting in which they functioned.

**Volunteer motivations theme.**   Most participants reported a strong desire to help their community combat the COVID-19 pandemic. The effects of the pandemic had simultaneously suspended in-person classes, clinical training, and routine work, leaving participants feeling idle and powerless. The contact tracing program offered a chance to respond to the pandemic by applying their knowledge and skills as budding or established practitioners, teachers, or researchers in the health sciences. Some noted that the safety of being able work remotely made telephone contact tracing more attractive than other volunteer opportunities that required physical interaction.

*"As someone who's going into this field, I've always wanted to do outbreak investigation and outbreak response kind of stuff. And it was really hard to feel powerless. And so, when this opportunity came up, I was like, this is something that I can do, like using my education and my skills." (Participant 4:1, Contact Notifier)*

*"I heard about this [volunteer opportunity] and how it could really use the skill sets of people who are trained to interview patients. And I thought that was one of the best ways I could help." (Participant 1:3, Case Investigator)*

**Time management theme.** Balancing contact tracing duties against other responsibilities was easy for some yet challenging for others. Unpredictable schedules, especially for students during examination periods, made it difficult for some to keep up with their tracing responsibilities. Case investigation volunteers were allowed to adjust their assigned case load every week to match their availability and used this flexibility to better manage their time and level of involvement. In contrast, contact notification volunteers were given fixed case assignments on a rotating basis and noted that their inability to control their workload could be stressful.

*"It's really hard to just get an email all of a sudden [that says] 'you have to call this person within 24 hours.' And I wasn't able to do all of them in the right timeframe because of that. I had too much other work to do." (Participant 6:2, Contact Notifier)*

In addition, participants from both teams reported that the sporadic workflow was challenging. Many said that they felt out of practice making calls after even a few days' hiatus, while others felt "burnt out" after making many calls in a short period.

*"I think the issue is just that the current inconsistency of not being as well-practiced in the ability to do the interview as well as you might have been doing them when you had a more consistent stream of calls." (Participant 2:1, Case Investigator)*

*"I found that if I was doing this for four hours a day, I got kind of pretty burned out by it. . . the next day I was like, I don't want to do this." (Participant 2:4, Case Investigator)*

**Knowledge theme.** Participants highlighted the importance of knowing and effectively applying current CDC COVID-19 recommendations and other policies regarding confidentiality and privacy protection. Because the tracing scripts changed frequently during the early phase of the pandemic to meet the changing CDC guidelines, participants found it challenging both to stay current and to feel confident that they were providing accurate guidance to cases or contacts in every situation.

*"[CDC guidelines] are very fuzzy and always evolving in terms of the recommendations. That makes it really hard to feel confident in what we're telling people." (Participant 2:2, Case Investigator)*

They also often found privacy protection policies difficult to apply due to the unique circumstances of each call. Participants said they were often required to speak with family members or other proxies in tense or emotional circumstances when cases or contacts were severely ill or unavailable to speak. A lack of familiarity with navigating and applying privacy policies within these unpredictable scenarios made it difficult for some participants to discern how much information they could appropriately share with proxies.

*"More information on [privacy and confidentiality], who we can and cannot tell certain things to, how to deal with proxies. I think that would have been really helpful." (Participant 3:3, Case Investigator)*

In addition, participants occasionally struggled to answer questions from cases about why certain data, such as information about contacts (name, phone number, age, gender) or current health status, needed to be collected. On the other hand, contacts frequently questioned where callers obtained such information. Participants believed that cases were hesitant to provide information when they were unsure what the city would do with it or thought that it would intrude on the privacy of their contacts. Although the importance and utility of each part of the script was covered in the volunteer training, participants felt that providing refresher sessions or opportunities to clarify how data would be used could have increased their ability to adequately answer case and contact questions in order to better promote faith and trust.

*"People would ask 'well, why do you need the information from my husband's phone number? He's here. He's listening to this conversation. He knows that I have COVID and that he's been exposed to COVID. So now I need to give the health department his phone number [so] that someone else can call him and tell him that I have COVID?' or, similarly, they'll say, 'Oh, everyone knows. I've told [my close contacts]. So why do I need to [give you their number]? They wouldn't want me giving [their phone number] so that you can bother them.'" (Participant 2:2; Case Investigator)*

*"I got a lot of 'where's this data [personal data about the individual] coming from, where are you getting this information?' And I think it would have been helpful to have known exactly where that information was coming from so that I could have a better answer for that." (Participant 6:2, Contact Notifier)*

**Skills theme.** All participants strongly agreed that effective communication skills were critical to establishing rapport with those facing a new and potentially frightening illness. Many adapted the interview script to this end. For example, rather than starting with standardized questions about demographics and specific symptoms, several participants found that beginning by asking open-ended questions about the client's current situation and setting expectations about content and length of the call helped engage some individuals.

*"I do a little bit more signposting in the interview than is included in the script. What I mean by that is setting an expectation about what are what are all the things we're going to talk about. . .I've found that sometimes cases are surprised by how long [the interview] is going, that they start to check out a little bit. Whereas if the expectation is very clear from the get-go, then I think people let their guard down a little bit and also just feel a little bit more comfortable with the interview." (Participant 1:5, Case Investigator)*

Applying techniques to communicate effectively and adapting messages in real-time was especially critical during moments of high emotion or conflict. For example, volunteers described unexpectedly being the first to inform cases about their positive test results or learning from those answering that the case had died or speaking to individuals who did not wish to participate. In one instance the participant had felt obligated to call 9-1-1 for emergency medical assistance for a case who was in physical distress. As discussed below, participants

repeatedly suggested that role-playing exercises be included in their training to build skills and confidence in navigating these emotionally charged or unanticipated situations.

*"In the training, I [would] definitely [add] like a role player, an initial call, for both a standard case and a nonstandard case. So, for instance, I've had a call where someone just started swearing at me on the phone, which is not something you expect and then I had another call where I had to call 9-1-1 on that person's behalf. . . So, I think, because we're calling sick people, getting [new trainees] ready for what a normal case looks like and what an abnormal case looks like, or just to get them able to think on their feet, would probably be important." (Participant 1:4, Case Investigator)*

**Collaborative learning theme.** Participants frequently mentioned a need for internal communication structures to better promote information exchange, shared learning, and timely adaptation to periodic changes in guidelines or programmatic priorities.

Having direct communications with program supervisors was very important to participants, and they especially appreciated quick responses to their questions. They felt it was important for supervisors to communicate changes in CDC guidelines and programmatic priorities, as well as to be responsive to volunteers' concerns, suggestions, and requests for changes or clarifications in protocols. These "two-way communications" were viewed as critical to maintaining "morale and faith" in the mission and promoting team solidarity. They described the chat application GroupMe as being "so helpful" in providing a direct mechanism to reach the supervisors with questions and view other volunteers' comments. However, several participants also noted that discussion threads were often basic and repetitive, reducing their value.

*"[The leaders] have been really responsive if I send the GroupMe message. People are pretty happy to respond, and they do that very quickly. That's been good." (Participant 3:2, Case Investigator)*

*"I am on the GroupMe, I've been on it for since I started, but I basically ignore it because there's so many messages that go into it and so many of them are, 'so I'm a new volunteer, how do I use [the interview software]?' And then it's a 20-message thread and everyone has to see it. So I check it like every few days and I just kind of scroll through." (Participant 3:6, Case Investigator)*

While the messaging app served as a useful hotline to request help from supervisors, many felt that other mechanisms were needed to provide peer-support, foster community, and learn from the experiences of their peers. Several mentioned that they thought that hearing about other volunteers' thoughts and experiences during the FGDs had been helpful. They advocated establishing a "buddy system" or regular meetings with small groups of peers for volunteers to share experiences, learn from one another, and debrief after difficult calls. They felt that such meetings could help to provide regular updates on changing protocols as well as promote a sense of community in an environment where in-person interactions were impossible.

*"At the moment I feel very comfortable doing a few practice calls with someone who's just starting, going through some of the situations I've been through, a regular one, a few difficult ones and literally spending 20 minutes, half an hour doing those things. If someone had done that for me at the start, I would've felt a lot more comfortable than I originally did." (Participant 5:1, Contact Notifier)*

"*I don't think that I necessarily need more training, but the GroupMe, [or] having a short meeting once a week with small groups to discuss newer things I think would be helpful.*" (Participant 4:2, Contact Notifier)

## Implementation dimension

Two themes emerged that concerned the *implementation* dimension: Tools and External Coordination. Each focused on key aspects of the feasibility of implementing a volunteer-based contact tracing program.

**Tools theme.**   Participants described several tools that were essential to their tasks, and the one most frequently discussed was the script. Many thought it was difficult to deliver the words verbatim because doing so made them sound "like a robot." As previously noted, many adapted the script language or individualized their introductions in an attempt to rectify this problem. However, when asked if replacing the script with a bulleted list of objectives would be preferable, most said that a word-for-word script was necessary as a training aid, especially during a tracer's first few calls. Other useful tools that participants regularly consulted included guidance documents provided by the program and a list of frequently asked questions. Participants appeared enthusiastic to add to and update these materials based on their own experiences. Some even suggested compiling examples of challenging call scenarios and response strategies into a reference document.

"*I don't think [any] amount of training can really prepare [you] for that first call. I know that sounds, I mean, yes, there was training to prepare for the first call, but I think you'd need that script just as that safety net.*" (Participant 5:2, Contact Notifier)

"*It would be helpful to have a repository of what these possible [call scenarios] are based on experience, real experience, and we could all contribute to that.*" (Participant 1:2, Case Investigator)

Participants also identified a need to adapt scripts and protocols for asymptomatic cases, minors, non-English speakers, and cases residing in congregate settings such as nursing homes. Calls to individuals in congregate settings were especially challenging to navigate because cases were often severely ill or otherwise incapacitated, caseloads were high, staff were already overwhelmed and overloaded with calls, and other factors. These cases were redirected to the Health Department for follow-up.

In addition to these specific contexts, participants felt that the contact notification process should be modified for members of a case's household and offered examples to support this suggestion. First, cases were often reluctant to provide information about their family members, possibly due to mistrust of the caller, fatigue and annoyance from being called by multiple agencies, or a desire to prevent additional calls to their household contacts. Second, even when this information was successfully collected by case investigators, they stated that it could be "a bit of a puzzle" to correctly identify cases and contacts at the beginning of a call without compromising privacy. When incorrect assumptions were inadvertently made or participants found themselves duplicating a call because a case had given the same phone number for multiple contacts (*e.g.*, phones shared within households), participants on the contact notification team said the calls felt like "a mess" and that they lost credibility with the contacts. Last, participants often struggled to identify a single exposure date for contacts who were living with a case and hence continuously exposed, whereas it was simpler to identify a discrete date for non-household contacts. All these experiences led volunteers to recommend conducting household contact notification together with the case interview.

*"There should be a separate [protocol] for household contacts. A household contact represents an ongoing exposure dynamic that is different than non-household contacts. If possible, the person calling the cases should contact the household contacts."* (Participant 4:2, Contact Notifier)

Data entry software was also identified as a critical contact tracing tool. Many case investigator participants appreciated the flexibility of New Haven's emergency-management software, as it allowed them to scroll through the questions and enter data in a smooth yet flexible order. This feature was highlighted as important when interviews did not follow the planned order of questions. With regard to data management, however, the free-text data collection tools used by the contact notification team were challenging to process, and several participants responsible for entering these data felt that they "should be updated" and recommended the use of a standardized form. Another tool that volunteers discovered to be helpful was dialer software (Doximity, San Francisco, CA) to mask their personal information, display the Health Department's phone number or a leave a virtual callback number (Google Voice, Mountain View, CA).

*"So, the number they see on their caller I.D. is from the [health department]. . . And then the number we leave [for] voicemail is my Google Voice number that will forward to my cell phone. So, they don't know who we are, they don't have our name or our personal information any more than what we say in the voicemail."* (Participant 1:4, Case Investigator)

**External coordination theme.**   Participants from both teams spoke often about how the activities of external organizations affected their own activities. Cases and contacts often reported receiving numerous calls from various organizations, such as healthcare providers, testing facilities, and insurance companies, and were often "annoyed" at hearing the "same things from multiple different people." In some instances, different organizations provided conflicting advice about isolation and quarantine periods. Ultimately, these experiences often led to call recipients being less receptive to engaging with the caller, thereby making it more difficult to collect the necessary information.

*"When there are multiple people who are giving [the case] recommendations that are not the same, it becomes challenging to feel like you are authoritative and for people to feel like they know what's going on."* (Participant 2:4, Case Investigator)

*"Because they've tested positive, their doctor has given them a lot of recommendations already and they're hearing it from us again. . .I think lately I've been getting more people who've tested positive and they've been annoyed with my call, more so I think because they've heard kind of the same things from multiple different people."* (Participant 5:4, Contact Notifier)

## Maintenance dimension

The final theme, Sustainability, focused on long-term threats to the volunteer-driven contact tracing program and aligned well with the *maintenance* dimension of RE-AIM.

**Sustainability theme.**   Several participants considered a volunteer-driven workforce as ideal for the "crisis phase" of an epidemic, allowing an accelerated response without the delays inherent in formal hiring or when re-assigning existing employees to contact tracing was not possible. However, participants did not see a volunteer-driven program as sustainable during the "maintenance phase" following the crisis. Some participants had recently graduated and

were departing for jobs or further training, and others planned to soon return to class or to other responsibilities.

> "*I think that you just unfortunately have to account for the fact that [volunteers] who are trained might be gone two months later and then [training and volunteer turnover] keeps going on and on again.*" (Participant 6:6, Contact Notifier)

As nearby states began hiring and paying contact tracers, other volunteers reported feeling frustrated, underappreciated, or less inclined to continue with the program. Participants suggested several strategies to maintain long-term involvement of contact tracers such as hiring them into part- or full-time paid positions or incentivizing student volunteers by offering academic credit for their work.

> "*I think it's a great idea for an acute crisis emergency for those first few weeks. Now this is an ongoing thing, I feel putting in a long-term solution, and maybe this counts as a practicum for the incoming students or for the continuing students or finding a way you can either weave this into the program. . . [or] They should be paying us to do it, not exorbitant amounts. But like 10 or 12 dollars an hour. . .I think that not doing that is a real disservice.*" (Participant 5:1, Contact Notifier)

### Synthesis of barriers, facilitators, and proposed solutions across themes and RE-AIM dimensions

Table 2 summarizes the facilitators, barriers, and potential solutions for improving implementation of contact tracing as reported by participants, mapped to their respective themes, and synthesized within the RE-AIM dimensions.

## Discussion

This is among the first studies to comprehensively describe the implementation context of COVID-19 contact tracing and provides a unique window into the rich experiences and perspectives of volunteers involved in a high-volume program at the peak of the April-June 2020 surge in the northeastern US. We identified many barriers to delivering this complex intervention in the midst of a public health emergency, but also several facilitators and many potential solutions for improving implementation, both in general and in the context of a volunteer program. Many insights echo the prior literature on contact tracing for other diseases, while others remain unique to the context of COVID-19 and the crisis scenario of a rapidly emerging pandemic. Categorizing our findings according to the RE-AIM framework allowed us to group many disparate themes into discrete, well-validated dimensions for improving implementation [21, 22].

The *reach* and *effectiveness* of COVID-19 contact tracing vary across settings, with proportions of cases successfully interviewed ranging from 53% - 99% [29, 30] and adherence to self-isolation instructions reported as low as 25% [30]. While the specific mechanisms driving these outcomes are not yet fully understood in the context of COVID-19, challenges to the *reach* and *effectiveness* of contact tracing in other settings have been associated with several client factors including anticipated stigma and loss of privacy [31–34], language barriers [31], and low public awareness of the importance of contact tracing [31, 33, 35, 36]. The apparent reluctance to answer our participants' calls may relate to several of these barriers. The RE-AIM framework suggests that COVID-19 contact tracing programs might consider engagement

**Table 2. Summary of findings organized by themes within the RE-AIM dimensions.**

| Dimensions | Themes | Facilitators | Barriers | Potential Solutions |
|---|---|---|---|---|
| Reach | Making Contact | Dialer software used to replace caller's personal phone number with a health department number | Low answer rate | Introduce text messages to introduce phone calls; obtain outreach preferences at testing |
| | Establishing Rapport | Dialer software used to replace caller's personal phone number with a health department number | Lack of trust in an unknown caller | Routinely address privacy concerns |
| | | Many cases and contacts willing to participate out of a desire to help their community | Low public unawareness of contact tracing leading to lack of interest or comfort in providing information about contacts | Organize public awareness campaigns; provide thorough explanations for why contact tracing is important for the community |
| Effectiveness | Delays | -- | Late reporting of test results | Automate test reporting and transfer of information to contact tracers |
| | | | Unknown language preferences | Verify language preferences at point-of-testing |
| | Community Needs | Health department routinely assesses needs as part of outreach | Lack of money, or adequate food & housing to help cases to adhere to isolation & quarantine | Increase funding for financial, nutritional, and housing supports; better inform tracers about how such needs can be met |
| Adoption | Volunteer Motivations | Partnerships with academic institutions and students | -- | Reward non-employed tracers with academic credit or certificates of experience |
| | Time Management | Weekly availability survey used for case investigation team | Shifting volunteer availability | Offer flexible, volunteer-driven scheduling |
| | | | Inconsistent workload due to varying case incidence with skill loss from inactivity | Ensure consistent baseline involvement with longitudinal skill refreshers |
| | Knowledge | Brief, targeted training provided to new volunteer tracers | Need for broad mastery of diverse content areas including biology, guidelines, procedures | Offer self-directed, online training modules to obtain baseline and knowledge |
| | | Many volunteers had previous education or experiences in health sciences | Frequent changes to guidelines due to evolving understanding of COVID-19 transmission dynamics | Frequently revise protocols to reflect changing guidelines, and rapidly communicate of these changes to the tracers; provide repository of potential call scenarios for outreach workers to learn from. |
| | Skills | Many volunteers previously trained in patient communication skills | Need for effective communication skills for building rapport | Incorporate role-plays and simulations to build up communication skills during training |
| | Collaborative Learning | Leaders regularly responded to questions by e-mail or GroupMe* | Lack of communication with leadership and feedback to ensure quality performance | Integrate two-way communication via messaging apps, email, and supervisory support |
| | | | Sense of isolation and lack of community while working remotely | Encourage peer mentorship, buddy systems, and regular, small-group peer meetings |
| Implementation | Tools | Software was flexible and allowed case investigators to adapt it to the interview at-hand. | Impersonal, non-conversational script | Personalize script and allow for adaptation to the clients' needs. |
| | | | Lack of interoperability of electronic systems | Provide simple and standardized data collection tools |
| | | Health department adapted script according to volunteer suggestions | Loss of volunteer privacy | Offer and/or require use of call masking software |
| | | | Need for specialized protocols for key populations† | Develop and apply specialized protocols |
| | External Coordination | -- | Duplicate calls to the same cases or contacts, leading to frustration and decreased engagement | Coordination with other clinics, laboratories, and health organizations to streamline and integrate communication |
| Maintenance | Sustainability | -- | High volunteer turnover; decreasing motivation over time | Offer payment or other compensation and acknowledgement such as academic credit or certificates of experience |

*Mobile app for hosting chat-groups

†Asymptomatic cases, residents of congregate settings, minors, non-English speakers, household contact.

strategies to enhance uptake such as using the initial point-of-testing interaction to identify optimal times to call and to document language preferences, and possibly using text messaging to identify and introduce outreach workers prior to calling. Our participants also highlighted the role of financial, nutritional, and social supports for those expecting or disclosing difficulties with isolation or quarantine as another way to potentially enhance the impact of contact tracing. Similar supports are commonly provided to tuberculosis cases to enhance patient outcomes and acceptance of contact tracing [37]. This notion is further reinforced by a recent anonymous survey study conducted in the UK which found that increased adherence to COVID-19 self-isolation and lockdown instructions was associated with having received help from anybody outside of the household [30].

A shortage of human resources is a major challenge to *adoption* of contact tracing for COVID-19 and other communicable diseases [31, 33, 38–41], both because many contact tracers are needed and because this capacity must be flexible enough to expand and contract with the waves and surges of the epidemic. In addition, as noted above, outreach workers must have good communication skills and a detailed knowledge of program policies and guidelines [35, 38, 42, 43]. Engaging/hiring volunteers is one option for rapidly scaling a pandemic contact tracing workforce [44] and was a strength identified by our study participants. There are also several personal benefits that might be highlighted to attract volunteers to such a workforce, including the anticipated satisfaction of contributing to the pandemic response and opportunity to gain practical experience in a health science field. Yet several challenges to a volunteer-driven workforce remain, such as managing shifting schedules and training lay persons from diverse backgrounds to act as public health agents. Allowing volunteers some degree of flexibility in their scheduling may allow programs to accommodate volunteers' external responsibilities, while using self-directed, online training modules [45] would decrease the initial training burden on local programs and allow them to focus their efforts in this area on ongoing education about local guidelines and practices, and on skill-building exercises such as role plays.

Another important insight from our participants about *adoption* of a volunteer contact tracing model included their suggestion to create a learning community to help them overcome their relative inexperience with outreach work. In other settings, pairing new trainees with those who are more experienced and/or facilitating an environment in which trainees can learn alongside their peers and support one another has been shown to increase trainee confidence and skill [46]. This sense of community seemed particularly important in the context of COVID-19 when requirements to work remotely made it more difficult to learn new skills because it was harder to receive feedback from peers or supervisors. We strongly recommend that COVID-19 contact tracing programs develop and promote robust communication and support structures within their organizations, using strategies such as peer-mentorship and regular, small-group meetings.

Within the *implementation* dimension, we found that properly designed tools for data collection and storage, specialized protocols for key populations, and coordination with external organizations were thought to be critical to success. These implementation factors may also have positive spillover effects for adoption, reach, and effectiveness in that efficient, user-friendly, non-redundant systems benefit call recipients and contact tracers alike. Two simple suggestions for improving efficiency included adopting more accessible tools for data collection and management, as shown with contact tracing for other diseases [33, 38, 39, 41, 47], and coordinating case and contact interviews within the same household as is commonly done in tuberculosis contact investigation [48, 49]. A threat that participants identified, poor interagency communication, has also been described during tuberculosis contact investigation in border regions [39]. In contrast, close coordination of Ebola contact tracing teams led to faster and greater uptake in the target populations [33, 50]. Further benefit was realized by these

response teams when they integrated services across disciplines, including social supports for basic needs and mental health, information-sharing with local community leaders, and public health interventions including active case-finding and quarantine. The experiences of our participants, combined with evidence from other contact tracing studies from contexts beyond COVID-19, emphasize the importance of coordinated, multidimensional outreach and support of cases and contacts.

Lastly, our study raises concerns about the *maintenance* of volunteer-driven contact tracing programs, with particular regard to sustainability. Our study participants noted that the initial motivations to volunteer out of altruism and/or a desire for practical experience can wane over time, particularly when neighboring programs began hiring full-time contact tracers. We found that volunteer availability can also change over time, especially for students and those under-employed as a consequence of physical lockdowns. These factors can make it difficult to establish and maintain a stable workforce. Payment or other forms of reward have been shown to increase motivation and commitment of tracers in other settings [31, 35, 38, 51], and our participants echoed the importance of feeling valued and appreciated for their efforts. We recommend that programs unable to hire employees for contact tracing consider providing academic credit or certificates to volunteers to acknowledge their critical contributions to pandemic response.

## Strengths and limitations

A key strength of this study is its timeliness in providing insight into how to respond more effectively to an ongoing, global pandemic. This is the first qualitative evaluation we know of for COVID-19 contact tracing, despite the wealth of media attention devoted to this topic. Obtaining such direct feedback from key stakeholders in the COVID-19 crisis is critical for understanding the complexities of implementation. Second, the use of an established implementation framework adds strength and clarity to our findings and eases interpretability for broader contexts. Third, volunteer contact tracing is a feasible and adaptable solution to COVID-19 contact tracing, and this article provides several strategic recommendations specific to volunteer-driven programs that may increase effectiveness and efficiency. Fourth, the participants in this study were all experienced in health care or public health settings and as such were able to reflect deeply on their experiences and provide specific recommendations. Many of the recommended solutions to challenges were swiftly incorporated into practice by the Health Department, and future studies may evaluate the impact of these changes on contact tracing outcomes. Last, while video conferencing platforms are typically not used to conduct FGDs, this study demonstrates that this methodological approach is acceptable to participants and feasible, except for occasional reductions in audio quality. Those using this technology should provide written and verbal instructions to participants on best practices to optimize audio quality and maintain courtesy during the sessions.

There are several study limitations to note. First, participants' responses may have been influenced by group dynamics or social desirability bias. To partially compensate for this limitation, our analysis incorporated comments from a follow-up survey of participants soliciting additional comments that they might have felt uncomfortable sharing in a group setting, or simply forgotten to mention. Second, insights about barriers and facilitators of a volunteer-driven program may not apply to a professional-driven contact tracing program. Similarly, our experience with health sciences students may not be generalizable to other volunteer groups. However, several findings likely apply to other types of contact tracing programs, including strategies for reaching and engaging cases and contacts, the importance of adapting protocols and support systems to the needs of the local community, and the potential value in

communication and coordination among different health agencies. Third, our findings include only the perspectives of volunteer tracers and not those of cases and contacts which will be explored in a subsequent analysis. Finally, while the insights and suggestions of the participants were used to modify the program, we unfortunately were not able evaluate their impact. With declining case numbers in Connecticut in June 2020, local health departments transitioned contact tracing responsibility to the state department of public health, and this program was discontinued.

## Conclusions

The unique experiences of the FGD participants highlight several strategies for improving volunteer-driven COVID-19 contact tracing programs, including adopting flexible approaches to training and scheduling volunteers and fostering networks to facilitate support and learning among volunteers. While a largely volunteer-driven contact tracing program was feasible and acceptable in the context of a public health crisis, its greatest challenge was achieving sustainability after the initial case surge. Despite the difficulties of implementing COVID-19 contact tracing, our findings suggest that a workforce that is well-capacitated, networked with its surrounding organizations, and able to adapt its services to the unique needs of its clients can overcome many of these challenges.

## Supporting information

**S1 Table. CO-REQ reporting guidelines.**
(DOCX)

**S1 Text. Semi-structured focus group discussion guide.**
(DOCX)

## Acknowledgments

We thank the many volunteers for their pivotal role in the local response to the COVID-19 pandemic. In a time of uncertainty and great need, they devoted their time and energy to contact tracing. Their tireless efforts demonstrate their altruism and passion for the wellbeing of their community. We also thank the New Haven Health Department for their partnership and creativity throughout this process, without which none of this work would have been possible.

## Author Contributions

**Conceptualization:** Tyler Shelby, Brian Weeks, Maritza Bond, Linda Niccolai, J. Lucian Davis, Lauretta E. Grau.

**Data curation:** Tyler Shelby, Rachel Hennein, Christopher Schenck, Katie Clark, Amanda J. Meyer, Justin Goodwin, Linda Niccolai.

**Formal analysis:** Tyler Shelby, Rachel Hennein, J. Lucian Davis, Lauretta E. Grau.

**Funding acquisition:** J. Lucian Davis.

**Investigation:** Tyler Shelby, Katie Clark, Lauretta E. Grau.

**Project administration:** Maritza Bond.

**Resources:** J. Lucian Davis.

**Supervision:** Brian Weeks, Maritza Bond.

**Writing – original draft:** Tyler Shelby, Rachel Hennein.

**Writing – review & editing:** Rachel Hennein, Christopher Schenck, Katie Clark, Amanda J. Meyer, Justin Goodwin, Brian Weeks, Maritza Bond, Linda Niccolai, J. Lucian Davis, Lauretta E. Grau.

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
