## [Decision Letter · Decision Letter 0]

9 Feb 2021

PONE-D-20-29667

Implementation of a Volunteer Contact Tracing Program for COVID-19 in the United States: A Qualitative Focus Group Study

PLOS ONE

Dear Dr. Davis,

Thank you for submitting your manuscript to PLOS ONE. After careful consideration, we feel that it has merit but does not fully meet PLOS ONE’s publication criteria as it currently stands. Therefore, we invite you to submit a revised version of the manuscript that addresses the points raised during the review process.

Please respond to each recommendation from both reviewers.

We look forward to receiving your revised manuscript.

Kind regards,

Jennifer Yourkavitch

Academic Editor

PLOS ONE

Journal Requirements:

2. Please provide additional details regarding participant consent. In the ethics statement in the Methods and online submission information, please ensure that you have specified how verbal consent was documented and witnessed.

3. Please provide the interview guide used as supporting information.

4.We note that you have indicated that data from this study are available upon request. PLOS only allows data to be available upon request if there are legal or ethical restrictions on sharing data publicly. For information on unacceptable data access restrictions, please see http://journals.plos.org/plosone/s/data-availability#loc-unacceptable-data-access-restrictions.

5.Thank you for stating the following in the Competing Interests section:

"JLD and TS declare a contract with the state of Connecticut to assist with the state's contact tracing program. "

Reviewers' comments:

Reviewer's Responses to Questions

**Comments to the Author**

1. Is the manuscript technically sound, and do the data support the conclusions?

Reviewer #1: Yes

Reviewer #2: Yes

2. Has the statistical analysis been performed appropriately and rigorously? 

Reviewer #1: N/A

Reviewer #2: N/A

3. Have the authors made all data underlying the findings in their manuscript fully available?

Reviewer #1: No

Reviewer #2: No

4. Is the manuscript presented in an intelligible fashion and written in standard English?

Reviewer #1: Yes

Reviewer #2: Yes

5. Review Comments to the Author

Reviewer #1: This manuscript was a pleasure to read and provided valuable insights into facilitators and barriers to conventional COVID-19 tracing in a volunteer program. The organization of results by theme, RE-AIM dimension, and facilitators/barriers/solutions is particularly helpful to allow the reader to digest the examples from study focus groups. There are two areas in particular in which the authors may improve this manuscript: 1) providing more detail about the program and its processes, and 2) incorporating more COVID-19 related literature into the discussion. Suggestions regarding these two topics, as well as some additional minor comments, are listed below.

Background:

A brief introduction to the benefits and/or appropriate use of conventional contact tracing, as compared to digital contact tracing, may be helpful to place this research in the context of the larger literature regarding contact tracing for COVID-19.

Please note typo: In the absence of a vaccine (3).

Methods:

Please provide some additional information about how the volunteer contact tracing program operated:

What was the source and process through which clients (i.e., positive cases) were brought to the attention of the contact tracing program?

How were changes in guidelines/recommendations communicated to volunteers over the course of the program?

Please describe the needs assessment process and the actions taken by contact tracers in response to identified needs.

Please clarify the number of eligible volunteers for the study. The authors mention 190 volunteers in the program, but only 142 available participants for the focus groups. Did these available participants exclude the 40 public health nurses added to the program in April (it does seem so since all participants reported a university affiliation in Table 1)? If so, why? Also, how did the authors choose the number of volunteers to exclude due to low case load?

Please clarify the consent process. The authors’ statement, “After obtaining verbal consent, the session recording was transcribed using an automated transcription service,” makes it sound as if consent was provided after the focus group was concluded. Please reword this statement to make it clear that participants provided consent prior to the start of the focus groups.

The authors state the use of inductive content analysis, but also categorized their findings based on deductive categorization surrounding facilitators/barriers/solutions (derived from the interview guide) and RE-AIM dimensions. Can the authors please elaborate on the use of both inductive and deductive coding and their coding process as it relates to the categorization of themes (presumably inductive), RE-AIM dimensions (presumably deductive), and facilitators/barriers/solutions (presumably deductive). What was the order in which these categorizations were made during data analysis?

Please note typo: we used the RE-AIM framework(21)

Results:

It would be helpful to split up Table 1 into the two study groups–case interviewers and contact notifiers–to assess differences in the study population based on role.

The free-text portion of the survey provided data in addition to the focus group transcripts. Were significant, unique findings revealed in the free-text responses that were not found in the focus study discussions?

For the adoption dimension, the authors state that information about “volunteers’ experiences” were allocated to this category. This seems a vague description, as all responses of the volunteers will reflect the perspectives of their experiences. Is there another way to describe the allocation of comments to the adoption dimension that provides greater specificity?

The authors state delays were at times related to “cases seeking testing.” It is not clear what this means in the context of the examples provided. Can the authors elaborate on this statement to communicate what this example is describing?

Please define “FGD”.

Discussion:

As Table 4 represents a synthesis of analyzed data it seems more appropriate to include it in the Results section, with additional statements about the identification of these additional themes (facilitators, barriers, solutions) in the text. Some of the information in this table is new information, not previously mentioned in the results analysis.

In Table 4, should the “Establishing rapport” label be merged with the empty cell below to denote its relevance to examples in both rows for this theme?

In Table 4, can “Automate data transfer” be more specific? Automate the communication of testing results?

While the discussion uses examples from contact tracing of other diseases, this section appears to lack references to citations regarding the proposal, implementation, or evaluation of contact tracing for COVID-19. It would be useful to include literature that can place the findings of this study in the context of what others have learned about the implementation, impact, or efficacy of contact tracing specific to COVID-19.

References:

Some reference information appears to be incomplete.

Reviewer #2: This article presents the findings from a timely study on the effectiveness of a volunteer contact tracing program for COVID-19. The study provides useful, concrete findings about constraints and how similar programs can be improved. The manuscript is clearly written and well organized.

Main comments:

• Pg. 10 on RE-AIM framework – It would be helpful to give more information about the RE-AIM framework and why you chose to use it since readers may not be familiar with it. I think the framework makes sense to use, but it is not well explained up front and is only explained in pieces in the results section. Please define each of the domains and how they are applicable to the contact tracing program. This will provide a stronger grounding for the data collected, the results, and how you will use the findings to make program recommendations.

• Pg. 26 table 4 – A lack of compensation was not discussed in the results section. Please include to support what is in the table or revise the table.

Other comments:

• Pg 4. Contributions to Science box – Note that these findings are about a specific context and the extent to which these findings may or may not be applicable to other parts of the US (e.g., where there is greater skepticism around COVID-19 for instance than in CT) or other countries (e.g., what types of similarities in the public health system would make the results applicable).

• Pg. 8 – Are there any methodological notes about conducting the focus groups over Zoom? Would be worth stating if no major challenges were faced at least since it’s newer to do over Zoom.

• Pg. 8 last full sentence – Edit for clarity.

• Pg. 8 end – May want to refer to the guide as a focus group guide rather than interview guide for clarity.

• Pg. 9 paragraph on transcripts – Edit paragraph for flow.

• Pg. 9 on saturation – Did you assess saturation for the case volunteers and contact volunteers separately? Also edit the sentence for clarity (e.g., iteratively reviewed the transcripts and conducted additional focus group discussions until no new themes emerged).

• Pg. 9 on ATLAS.ti – The current description sounds like you coded the data and then entered the data into ATLAS.ti rather than coding the data in ATLAS.ti. If you did code the data on paper or in some other program, specify that and clarify what you mean by “entering the data” into ATLAS.ti.

• Pg. 10 on study sample – Why did you select the first people who responded instead of using other purposive criteria? Do you know how the demographic characteristics of your sample compare to broader group of volunteers?

• Pg. 11 after table – Missing header for shift to discussing identified themes.

• Pg. 13-14 table 3 – It would be helpful to split up the table and include in each dimension so you can see the table with the results for that dimension. The content of the table is useful, but it is a bit out of context to see the themes and quotes before the presentation of the results for the dimension. Splitting it up would also allow you to easily see the themes for the domain and reference illustrative quotes.

• Pg. 14 table 3 – I am not sure I understand the purpose of the legend text. It doesn’t seem needed.

• Pg. 18 – Inconsistent use of FGD acronym.

• Pg. 20 – I wonder if a different label would make sense instead of community of practice. It seems that this theme is more about having supportive supervision and peer-learning/support. Community of practice is more often referred to network going across programs or organizations rather than communication within a program.

• Pg. 31 last paragraph – Would clarify that sending the survey afterwards would not mitigate social desirability bias, but may have given you some insight into potential sources of bias.

6. PLOS authors have the option to publish the peer review history of their article (what does this mean?). If published, this will include your full peer review and any attached files.

Reviewer #1: **Yes: **Sarah R. MacEwan

Reviewer #2: No

---

## [Author Response · Author response to Decision Letter 0]

6 Mar 2021

March 6, 2021

Dear Editors and Reviewers:

We would like to thank you for giving us the opportunity to revise and resubmit our manuscript entitled “Implementation of a Volunteer Contact Tracing Program for COVID-19 in the United States: A Qualitative Study.” 

We have addressed all the points mentioned by the editors and reviewers and have included our detailed, point-by-point responses below. We feel that your constructive feedback has greatly strengthened this article. Please do not hesitate to contact us if you need further information or clarification.

Sincerely,

J. Lucian (Luke) Davis, MD, MAS, Corresponding Author

 

Editor Comments and Responses

a. Response: We have reviewed PLOS ONE’s style/formatting requirements and made several changes, including:

i. Reducing abstract word count to below the limit

ii. Removed footnotes from the Manuscript 

iii. Added a Short Title to the Manuscript’s Title Page

iv. Added a new Methods sub-section detailing IRB approval and consent procedures. 

2. Please provide additional details regarding participant consent. In the ethics statement in the Methods and online submission information, please ensure that you have specified how verbal consent was documented and witnessed.

a. Response: Additional details and clarifications regarding consent have been added to the “Ethics Statement and Consent Procedures” section of the Methods: 

i. Page 11, Line 213: “The study protocol was approved by the Yale Human Subjects Committee (Institutional Review Board Panel A for Social, Behavioral, and Educational Research) and the New Haven Health Department. A waiver of written consent was approved by the Human Subjects Committee because the study posed no greater than minimal risk and did not involve any procedures that would require written consent in a non-research context. Before video-recording the session, the group facilitators read the consent form aloud and obtained verbal consent from all participants to participate in the study and be recorded.”

3. Please provide the interview guide used as supporting information.

a. Response: The guide has now been provided as “Supplemental Text 1.”

i. If there are ethical or legal restrictions on sharing a de-identified data set, please explain them in detail (e.g., data contain potentially identifying or sensitive patient information) and who has imposed them (e.g., an ethics committee). Please also provide contact information for a data access committee, ethics committee, or other institutional body to which data requests may be sent.

ii. If there are no restrictions, please upload the minimal anonymized data set necessary to replicate your study findings as either Supporting Information files or to a stable, public repository and provide us with the relevant URLs, DOIs, or accession numbers. Please see http://www.bmj.com/content/340/bmj.c181.long for guidelines on how to de-identify and prepare clinical data for publication. For a list of acceptable repositories, please see http://journals.plos.org/plosone/s/data-availability#loc-recommended-repositories.

iii. We will update your Data Availability statement on your behalf to reflect the information you provide.

1. Response: Regarding the request to deposit source data in an online repository, there are a number of ethical and legal reasons which prevent us from doing so. First, participants did not provide consent for the transcripts to be deposited publicly. Second, because the focus groups describe personal experiences, it is not possible to fully deidentify the transcripts, and there is a risk of loss of confidentiality for participants given the modest size of the contact tracing program and of the community. Third, the New Haven Health Department provides oversight for the data collected by and for its contact tracing program, and by policy requires any use of this data to be directly approved by the Health Department. For these reasons, data may only be made available upon request made to the Corresponding Author and the New Haven Health Department.

5. Thank you for stating the following in the Competing Interests section: "JLD and TS declare a contract with the state of Connecticut to assist with the state's contact tracing program. "

c. Please know it is PLOS ONE policy for corresponding authors to declare, on behalf of all authors, all potential competing interests for the purposes of transparency. PLOS defines a competing interest as anything that interferes with, or could reasonably be perceived as interfering with, the full and objective presentation, peer review, editorial decision-making, or publication of research or non-research articles submitted to one of the journals. Competing interests can be financial or non-financial, professional, or personal. Competing interests can arise in relationship to an organization or another person.

i. Response: Our new Competing Interests statement is as follows:

1. JLD and TS declare a contract with the state of Connecticut to assist with the state's contact tracing program. This does not alter our adherence to PLOS ONE policies on sharing data and materials.

a. Response: The ethics statement covering approval from Yale IRB and the New Haven Health Department, as well as consent processes, has been added to the Methods section, and is quote above in response to Editor Comment #2. 

a. Response: We have added captions for the two Supporting Information items uploaded (CO-REQ Checklist and FGD Guide)

 

Reviewer #1 Comments and Responses

General Comment: This manuscript was a pleasure to read and provided valuable insights into facilitators and barriers to conventional COVID-19 tracing in a volunteer program. The organization of results by theme, RE-AIM dimension, and facilitators/barriers/solutions is particularly helpful to allow the reader to digest the examples from study focus groups. There are two areas in particular in which the authors may improve this manuscript: 1) providing more detail about the program and its processes, and 2) incorporating more COVID-19 related literature into the discussion. Suggestions regarding these two topics, as well as some additional minor comments, are listed below.

Background:

1. A brief introduction to the benefits and/or appropriate use of conventional contact tracing, as compared to digital contact tracing, may be helpful to place this research in the context of the larger literature regarding contact tracing for COVID-19.

a. Response: Good point; we have added a summative sentence to the end of the first Background paragraph.

i. Page 5, Line 80: “Meanwhile, the lack of feasibility and acceptability of the best alternative, digital contact tracing, has ensured that person-led strategies will likely remain the first-line approach in most settings (19).”

2. Please note typo: In the absence of a vaccine (3).

a. Response: We have updated this sentence in the first Background paragraph to reflect changes in vaccine development/deployment and the typo is resolved.

Methods:

3. Please provide some additional information about how the volunteer contact tracing program operated:

i. What was the source and process through which clients (i.e., positive cases) were brought to the attention of the contact tracing program?

ii. How were changes in guidelines/recommendations communicated to volunteers over the course of the program?

iii. Please describe the needs assessment process and the actions taken by contact tracers in response to identified needs.

b. Response: Thank you for the suggestions. We added a new citation to an article we published in AJPH that describes the program itself in more detail, and also added several sentences to the “Setting and Procedures” section:

i. Page 7, Line 117: “The Health Department established a partnership with Yale University for volunteer contact tracing on March 27, 2020, as previously described (22).”

ii. Page 8, Line 137: “Each day, the Health Department’s lead epidemiologist identified new positive COVID-19 cases from the state’s reportable disease database and shared their corresponding outreach information with the case investigation team.”

iii. Page 8, Line 142: “The case investigator team shared a daily list of reported contacts, without any information regarding their respective cases, with the contact notification team via email.”

iv. Page 8, Line 146: “Case investigators routinely asked cases about food or housing insecurities, ability to isolate within homes, access to medical care, and other social needs, while providing numbers to local support organizations or free clinics when applicable. Contact notifiers also provided links to resources when applicable but did not routinely assess contacts for the same needs.”

v. Page 8, Line 150: “Team leads communicated changes in guidelines and protocols to volunteers via email and modified data collection forms appropriately.”

4. Please clarify the number of eligible volunteers for the study. The authors mention 190 volunteers in the program, but only 142 available participants for the focus groups. Did these available participants exclude the 40 public health nurses added to the program in April (it does seem so since all participants reported a university affiliation in Table 1)? If so, why? Also, how did the authors choose the number of volunteers to exclude due to low case load?

a. Response: Thank you for the suggested clarification. We have revised this section to increase clarity: 

i. Page 8, Line 155: “Eligibility criteria included being a volunteer in the case investigator or contact notification teams. We excluded the less experienced case investigators, defined as being in the lowest 25th percentile of total case assignments (<7 assignments). We did not exclude any contact notifiers because all assignments were distributed equally among this team, whereas case investigators were able to adjust their availability each week. We emailed invitations to all eligible volunteers to participate in the study. We set an initial recruitment goal of 18 participants from each team based on estimates of the number of focus groups required for thematic saturation (25).”

b. We excluded nurses given the focus of the manuscript on the experiences of new volunteers, as stated on Page 6, Line 101:

i. “Because volunteers were and still are key stakeholders in many contact tracing programs, learning about their experiences is vital for sustaining and scaling up contact tracing. To this end, we conducted focus group discussions (FGDs) with volunteers participating in a contact tracing program in Connecticut.”

5. Please clarify the consent process. The authors’ statement, “After obtaining verbal consent, the session recording was transcribed using an automated transcription service,” makes it sound as if consent was provided after the focus group was concluded. Please reword this statement to make it clear that participants provided consent prior to the start of the focus groups.

a. Response: Good point, we moved content regarding consent procedures to the Ethics Statement and Consent Procedures sub-section of the Methods and clarified. 

i. Page 11, Line 217: “Before video-recording the session, the group facilitators read the consent form aloud and obtained verbal consent from all participants to be in the study and be recorded.”

6. The authors state the use of inductive content analysis, but also categorized their findings based on deductive categorization surrounding facilitators/barriers/solutions (derived from the interview guide) and RE-AIM dimensions. Can the authors please elaborate on the use of both inductive and deductive coding and their coding process as it relates to the categorization of themes (presumably inductive), RE-AIM dimensions (presumably deductive), and facilitators/barriers/solutions (presumably deductive). What was the order in which these categorizations were made during data analysis?

i. Response: Valid points. We began with inductive codebook development and thematic analysis, followed by deductive categorization of themes according to RE-AIM dimension followed by identification of facilitators, barriers, and solutions within each RE-AIM dimension. These steps are now clarified as follows: 

1. Page 10, Line 192: “The coding team (TS, RH, LG) independently reviewed one case and one contact transcript and met to discuss and develop the codebook inductively.”

2. Page 10, Line 198: “the data were subsequently entered into ATLAS.ti (Version 8) and analyzed iteratively using thematic analysis (27)”

3. Page 10, Line 202: “After the themes had been identified, we used the RE-AIM framework (21, 28) to deductively organize emergent themes”

4. Page 11, Line 208: “Once organized according to the RE-AIM framework, we identified specific barriers, facilitators, and solutions within each RE-AIM dimension.”

7. Please note typo: we used the RE-AIM framework(21)

a. Response: Thank you for noticing this error; we have fixed the typo. 

Results:

8. It would be helpful to split up Table 1 into the two study groups–case interviewers and contact notifiers–to assess differences in the study population based on role.

a. Response: We appreciate this suggestion and have modified the table accordingly. Note that there was an incorrect labeling of one participant as “staff” rather than “student” in the prior table, which was now been corrected. 

9. The free-text portion of the survey provided data in addition to the focus group transcripts. Were significant, unique findings revealed in the free-text responses that were not found in the focus study discussions?

a. Response: This is an excellent point, and we have added a sentence to note that no differences in themes were noted across participant characteristics or between responses from the FGDs vs. survey responses: 

i. Page 13, Line 242: “There were no differences in themes expressed by volunteer type or by participant demographics or between the FGDs and follow-up free-text surveys.”

10. For the adoption dimension, the authors state that information about “volunteers’ experiences” were allocated to this category. This seems a vague description, as all responses of the volunteers will reflect the perspectives of their experiences. Is there another way to describe the allocation of comments to the adoption dimension that provides greater specificity?

a. Response: Thank you for the suggestion; we have adjusted the wording in the Methods section to more accurately describe what type of content was categorized within the Adoption dimension: 

i. Page 10, Line 205: “[themes about] volunteer delivery of the intervention and the setting in which they operated to the adoption dimension”

11. The authors state delays were at times related to “cases seeking testing.” It is not clear what this means in the context of the examples provided. Can the authors elaborate on this statement to communicate what this example is describing?

a. Response: We modified this sentence to increase the clarity of the message: 

i. Page 18, Line 286: “While some delays in the overall contact tracing process were beyond the control of the program, such as cases choosing to delay seeking COVID testing or slow reporting of test results, participants felt that identifying cases in need of translators before the first call was an actionable way to prevent additional delay.”

12. Please define “FGD”.

a. Response: We now define FGD in the last Background paragraph.

Discussion:

13. As Table 4 represents a synthesis of analyzed data it seems more appropriate to include it in the Results section, with additional statements about the identification of these additional themes (facilitators, barriers, solutions) in the text. Some of the information in this table is new information, not previously mentioned in the results analysis.

a. Response: We agree and have moved the table and a brief paragraph introducing it into the Results section.

14. In Table 4, should the “Establishing rapport” label be merged with the empty cell below to denote its relevance to examples in both rows for this theme?

a. Response: This is a good suggestion and we have made this correction.

15. In Table 4, can “Automate data transfer” be more specific? Automate the communication of testing results?

a. Response: We adjusted the wording to be more specific per the reviewer’s suggestion: 

i. Table 7: “Automate test reporting and transfer of information to contact tracers”

16. While the discussion uses examples from contact tracing of other diseases, this section appears to lack references to citations regarding the proposal, implementation, or evaluation of contact tracing for COVID-19. It would be useful to include literature that can place the findings of this study in the context of what others have learned about the implementation, impact, or efficacy of contact tracing specific to COVID-19.

a. Response: Thank you for this suggestion; we have added several new citations alongside adjusted wording in the Discussion to place our findings better within the context of COVID-19 when possible. Examples include: 

i. Page 34, Line 488: “The reach and effectiveness of COVID-19 contact tracing vary across settings, with proportions of cases successfully interviewed ranging from 53% - 99% (29, 30) and adherence to self-isolation instructions reported as low as 25% (30).”

ii. Page 35, Line 503: “This notion [need for support with quarantine] is further reinforced by a recent anonymous survey study conducted in the UK which found that increased adherence to COVID-19 self-isolation and lockdown instructions was associated with having received help from anybody outside of the household (30).”

iii. “Engaging/hiring volunteers is one option for rapidly scaling a pandemic contact tracing workforce (44)”

References:

17. Some reference information appears to be incomplete.

a. Response: We have reviewed the references and added updated several references to include their respective urls and access/citation dates. 

 

Reviewer #2 Comments and Responses

General Comment: This article presents the findings from a timely study on the effectiveness of a volunteer contact tracing program for COVID-19. The study provides useful, concrete findings about constraints and how similar programs can be improved. The manuscript is clearly written and well organized.

Main comments: 

1. Pg. 10 on RE-AIM framework – It would be helpful to give more information about the RE-AIM framework and why you chose to use it since readers may not be familiar with it. I think the framework makes sense to use, but it is not well explained up front and is only explained in pieces in the results section. Please define each of the domains and how they are applicable to the contact tracing program. This will provide a stronger grounding for the data collected, the results, and how you will use the findings to make program recommendations.

a. Response: Good point. We added additional content and a new reference in the Background following the first mention of RE-AIM in order to define each dimension: 

i. Page 6, Line 95: “The RE-AIM framework has been employed extensively for this purpose (21) and contains five dimensions: (1) reach, which focuses on the population an intervention targets and the process of engaging them, (2) effectiveness, which focuses on the intended impact of an intervention and potential barriers to that impact, (3) adoption, which focuses on the setting and individuals delivering the intervention, (4) implementation, which focuses on intervention protocols and strategy, and (5) maintenance, which focuses on intervention sustainability and scalability.”

2. Pg. 26 table 4 – A lack of compensation was not discussed in the results section. Please include to support what is in the table or revise the table.

a. Response: We previously discussed lack of compensation in the Maintenance theme but recognize that you are correct in pointing out that Table 4 also mentions this as a barrier in the Adoption theme. We removed it from the Adoption theme so that Table 4 now aligns with the text and prior thematic categorizations.

Other comments: 

3. Pg 4. Contributions to Science box – Note that these findings are about a specific context and the extent to which these findings may or may not be applicable to other parts of the US (e.g., where there is greater skepticism around COVID-19 for instance than in CT) or other countries (e.g., what types of similarities in the public health system would make the results applicable).

a. Response: We have added an additional bullet-point to this box describing how the findings may apply beyond volunteer-driven programs, but likely not to all contact tracing programs due to differences in program practices and community engagement across different locations: 

i. Contributions to Science Box: “While many of the findings from this study likely apply beyond the context of volunteer-driven, phone-based contact tracing programs, they may not be wholly applicable to other parts of the US or world that feature differences in public health infrastructure, use of technology for contact tracing, or community skepticism regarding COVID-19.”

4. Pg. 8 – Are there any methodological notes about conducting the focus groups over Zoom? Would be worth stating if no major challenges were faced at least since it’s newer to do over Zoom.

a. Response: Thank you for this suggestion; we have added new sentences in the Discussion: 

i. Page 38, Line 580: “Lastly, while video conferencing platforms are typically not used to conduct FGDs, this study demonstrates this methodological approach to be acceptable to participants and feasible, except for occasional reductions in audio quality. Those using this technology should provide written and verbal instructions to participants on best practices to optimize audio quality and courtesy during the sessions.”

5. Pg. 8 last full sentence – Edit for clarity.

a. Response: We edited the sentence to remove duplicate words and increase clarity. 

6. Pg. 8 end – May want to refer to the guide as a focus group guide rather than interview guide for clarity.

a. Response: Thank you, we have made the change in the text. 

7. Pg. 9 paragraph on transcripts – Edit paragraph for flow.

a. Response: We modified the beginning of this paragraph to move the consent material to its own section and switched from passive voice to active when appropriate. 

8. Pg. 9 on saturation – Did you assess saturation for the case volunteers and contact volunteers separately? Also edit the sentence for clarity (e.g., iteratively reviewed the transcripts and conducted additional focus group discussions until no new themes emerged).

a. Response: We modified this sentence to clarify: 

i. Page 10, Line 186: “Two moderators (TS and LG) iteratively assessed the content of case sessions until no new themes emerged (i.e., saturation had been reached), and separately followed the same process for contact sessions (25).”

9. Pg. 9 on ATLAS.ti – The current description sounds like you coded the data and then entered the data into ATLAS.ti rather than coding the data in ATLAS.ti. If you did code the data on paper or in some other program, specify that and clarify what you mean by “entering the data” into ATLAS.ti.

a. Response: We clarified in the revised draft that initial coding was done using Microsoft Word and subsequently transferred into ATLAS.ti.

i. Page 10, Line 197: “The coding team initially used Microsoft Word for coding, and the data were subsequently entered into ATLAS.ti (Version 8)”

10. Pg. 10 on study sample – Why did you select the first people who responded instead of using other purposive criteria? Do you know how the demographic characteristics of your sample compare to broader group of volunteers?

a. Response: Thank you for suggesting these clarifications. We updated the wording in the Methods to clarify our enrollment strategy and procedures.

i. Page 9, Line 159: “We emailed invitations to all eligible volunteers to participate in the study. We set an initial recruitment goal of 18 participants from each team based on estimates of the number of focus groups required for thematic saturation [25]. We enrolled participants consecutively until the target sample size was reached, ensuring balanced representation of volunteers from different schools and university positions (i.e. students, faculty, and staff).”

b. We also added a sentence regarding the representativeness of the study sample to the larger volunteer population. 

i. Page 12, Line 229: “School affiliations within the sample were similar to the those on the volunteer team overall, with a slightly lower representation of nursing students and a higher representation of faculty and staff in the study sample.”

11. Pg. 11 after table – Missing header for shift to discussing identified themes.

a. Response: We have added a heading per the reviewer’s suggestion. 

12. Pg. 13-14 table 3 – It would be helpful to split up the table and include in each dimension so you can see the table with the results for that dimension. The content of the table is useful, but it is a bit out of context to see the themes and quotes before the presentation of the results for the dimension. Splitting it up would also allow you to easily see the themes for the domain and reference illustrative quotes.

a. Response: We agree with this suggestion and have split both Tables 2 and 3 so that each RE-AIM dimension now has its own table. However, if the editors prefer a smaller number of tables we can recombine. 

13. Pg. 14 table 3 – I am not sure I understand the purpose of the legend text. It doesn’t seem needed.

a. Response: Upon revisiting the legend, we agree with the reviewer and have removed the legend as a similar statement appears elsewhere in the manuscript. 

14. Pg. 18 – Inconsistent use of FGD acronym.

a. Response: Thank you for attention to detail. We now define FGD in the Background and more consistently use the acronym throughout.

15. Pg. 20 – I wonder if a different label would make sense instead of community of practice. It seems that this theme is more about having supportive supervision and peer-learning/support. Community of practice is more often referred to network going across programs or organizations rather than communication within a program.

a. Response: This is a valid point. We agree that using a term that more specifically encompasses peer learning would be a better fit. Thus, we changed “community of practice” to “collaborative learning” given the participants’ emphasis that learning from more experienced volunteers and team leaders could facilitate new volunteers’ work. We revised the Discussion accordingly:

i. Page 36, Line 527: “In other settings, pairing new trainees with those more experienced and/or facilitating an environment in which trainees can learn alongside their peers and support one another has been shown to increase trainee confidence and skill (47).”

16. Pg. 31 last paragraph – Would clarify that sending the survey afterwards would not mitigate social desirability bias, but may have given you some insight into potential sources of bias.

a. Response: We added an additional sentence to clarify that the survey may have reduced bias due to the group dynamics but was less likely to have reduced social desirability bias due to participant perceptions of the research team: 

i. Page 38, Line 588: “The follow-up survey sent to each participant was intended to reduce social desirability unique to the group discussion context. However, we note that it may not have reduced social desirability bias when interacting with the group facilitators.”

---

## [Editor Report · Decision Letter 1]

22 Mar 2021

PONE-D-20-29667R1

Implementation of a Volunteer Contact Tracing Program for COVID-19 in the United States: A Qualitative Focus Group Study

PLOS ONE

Dear Dr. Davis,

Thank you for submitting your manuscript to PLOS ONE. After careful consideration, we feel that it has merit but does not fully meet PLOS ONE’s publication criteria as it currently stands. Therefore, we invite you to submit a revised version of the manuscript that addresses the points raised during the review process.

We look forward to receiving your revised manuscript.

Kind regards,

Jennifer Yourkavitch

Academic Editor

PLOS ONE

Journal Requirements:

Additional Editor Comments (if provided):

Thank you for addressing the reviewer comments. Please integrate the quotes from Tables 2 - 6 into the text in the Results section and eliminate those tables. Table 7 is a nice summary and should remain (renumbered as Table 2).

---

## [Author Response · Author response to Decision Letter 1]

8 Apr 2021

April 8, 2021

Editorial Board

PLoS ONE

Dear Editors:

We would like to thank you again for giving us the opportunity to revise and resubmit our manuscript entitled “Implementation of a Volunteer Contact Tracing Program for COVID-19 in the United States: A Qualitative Study, ” (PONE-D-20-29667R1).

We have now addressed all the points requested by the editors and have included our detailed, point-by-point responses below. We have also made very modest edits to the first and third sentences of the manuscript to better align with the current circumstances of the pandemic. Please do not hesitate to contact us if you need further information or clarification.

Sincerely,

J. Lucian (Luke) Davis, MD, MAS, Corresponding Author

 

Journal Requirements, Editor Comments and Responses 

(Note that Line Numbers reference the Tracked Changes version of the manuscript, with changes visible)

1. Please review your reference list to ensure that it is complete and correct. If you have cited papers that have been retracted, please include the rationale for doing so in the manuscript text or remove these references and replace them with relevant current references. Any changes to the reference list should be mentioned in the rebuttal letter that accompanies your revised manuscript. If you need to cite a retracted article, indicate the article’s retracted status in the References list and also include a citation and full reference for the retraction notice.

a. Response: Thank you for noticing this error. There were three references related to “Communities of Practice” literature that are no longer relevant following our previous revisions. The citations were previously removed from the text but remained in the reference list; we corrected this by removing them from the reference list and adjusting the citation numbers accordingly. 

2. Thank you for addressing the reviewer comments. Please integrate the quotes from Tables 2 - 6 into the text in the Results section and eliminate those tables. Table 7 is a nice summary and should remain (renumbered as Table 2).

a. Response: We moved all quotes from the table into the text within the relevant sections for each Theme and relabeled Table 7 as Table 2. In order to properly make the transition from table-to-text presentation of quotes, we made some minor adjustments to the text and added several new quotes that more directly reflect the content in the text. The additional changes are noted below:

i. Line 297: This quote was relocated from the Skills Theme to the Establishing Rapport Theme, as it more accurately captures the lack of prior work experience which is a topic in the prior paragraph (line 295). 

ii. Line 302: We added a new quote to reflect the importance of prior work experience which is a topic in the prior paragraph (line 294)

iii. Line 455: We edited this paragraph to better describe the previously included quote that reflects questions that contacts ask regarding the information that callers have. “On the other hand, contacts frequently questioned where callers obtained such information.”

iv. Line 457: We also added minor text changes to describe additional reasons why cases were hesitant to provide information about cases: “or thought that it would intrude on the privacy of their contacts.”

(a) Line 463: we added a new quote to reflect this finding. 

v. Line 500: We added a new quote to reflect the discussion of role plays noted in the prior paragraph (line 497)

vi. Line 527: We added a new quote to reflect some participants’ feelings that the GroupMe was repetitive, as described in the prior paragraph (line 521)

vii. Line 612: We added a minor in-paragraph quotation to describe how participants felt about the free-text data collection tools: “felt that they “should be updated” 

viii. Line 614: We updated the text about dialer software to accommodate the addition of an accompanying quote (line 618)

---

## [Editor Report · Decision Letter 2]

19 Apr 2021

Implementation of a Volunteer Contact Tracing Program for COVID-19 in the United States: A Qualitative Focus Group Study

PONE-D-20-29667R2

Dear Dr. Davis,

We’re pleased to inform you that your manuscript has been judged scientifically suitable for publication and will be formally accepted for publication once it meets all outstanding technical requirements.

Kind regards,

Jennifer Yourkavitch

Academic Editor

PLOS ONE
---

## [Editor Report · Acceptance letter]

28 Apr 2021

PONE-D-20-29667R2 

Implementation of a volunteer contact tracing program for COVID-19 in the United States: A qualitative focus group study 

Dear Dr. Davis:

I'm pleased to inform you that your manuscript has been deemed suitable for publication in PLOS ONE. Congratulations! Your manuscript is now with our production department. 

Kind regards, 

on behalf of

Dr. Jennifer Yourkavitch 

Academic Editor

PLOS ONE